# Autophagic Proteome in Two *Saccharomyces cerevisiae* Strains during Second Fermentation for Sparkling Wine Elaboration

**DOI:** 10.3390/microorganisms8040523

**Published:** 2020-04-06

**Authors:** Juan Antonio Porras-Agüera, Jaime Moreno-García, María del Carmen González-Jiménez, Juan Carlos Mauricio, Juan Moreno, Teresa García-Martínez

**Affiliations:** 1Department of Microbiology, Agrifood Campus of International Excellence ceiA3, C6 building, Campus de Rabanales, University of Córdoba, E-14014 Córdoba, Spain; b02poagj@uco.es (J.A.P.-A.); b62mogaj@uco.es (J.M.-G.); b02gojim@uco.es (M.d.C.G.-J.); mi2gamam@uco.es (T.G.-M.); 2Department of Agricultural Chemistry, Agrifood Campus of International Excellence ceiA3, C3 building, Campus de Rabanales, University of Córdoba, E-14014 Córdoba, Spain; qe1movij@uco.es

**Keywords:** sparkling wine, yeast, CO_2_ overpressure, protein, autophagy

## Abstract

A correlation between autophagy and autolysis has been proposed in order to accelerate the acquisition of wine organoleptic properties during sparkling wine elaboration. In this context, a proteomic analysis was carried out in two industrial *Saccharomyces cerevisiae* strains (P29, conventional sparkling wine strain and G1, implicated in sherry wine elaboration) with the aim of studying the autophagy-related proteome and comparing the effect of CO_2_ overpressure during sparkling wine elaboration. In general, a detrimental effect of pressure and second fermentation development on autophagy-related proteome was observed in both strains, although it was more pronounced in flor yeast strain G1. Proteins mainly involved in autophagy regulation and autophagosome formation in flor yeast G1, and those required for vesicle nucleation and expansion in P29 strain, highlighted in sealed bottle. Proteins Sec2 and Sec18 were detected 3-fold under pressure conditions in P29 and G1 strains, respectively. Moreover, ‘fingerprinting’ obtained from multivariate data analysis established differences in autophagy-related proteome between strains and conditions. Further research is needed to achieve more solid conclusions and design strategies to promote autophagy for an accelerated autolysis, thus reducing cost and time production, as well as acquisition of good organoleptic properties.

## 1. Introduction

Sparkling wines elaboration process (traditional method or ‘Champenoise’) involves a secondary fermentation in sealed bottle, followed by an aging period, at least 9 months for cava (a Spanish sparkling wine), where yeast cells must face several stress factors such as high ethanol concentrations (9.5–11.5 % *v*/*v*) and, above all, the endogenous CO_2_ overpressure, which reach values of 6–7 bar inside the bottle. The whole process including the fermentative process and aging is known as “prise de mousse” [1,2]. Along aging, the contact of wine with dying yeast cells leads to the release of their cellular content during a self-degradative process known as autolysis [3,4]. This event is fundamental for the quality of sparkling wines and improvement of the organoleptic properties.

Nevertheless, autolysis is a slow process and the development of strategies to accelerate this event has become an object of study for the enological industry [5,6]. These authors reported that autophagy takes place during secondary fermentation of sparkling wines and proposed the use of yeast strains with deregulated autophagy in order to accelerate the autolysis process. Autophagy is induced mainly under starvation conditions and involves the transport and degradation of cytoplasmic compounds in the vacuole [7,8]. It can be classified into two main types: macroautophagy and microautophagy, which are both selective and non-selective processes. Among selective autophagy, the Cvt pathway appears to be a biosynthetic process where hydrolytic enzymes are transported by double-membrane vesicles, smaller than autophagosomes, and delivered to the vacuole [9]. Both selective and non-selective autophagy share the molecular machinery, encoded by 33 autophagy-related (ATG) genes [10], of which 17 are required for both. Although autophagy has been commonly associated with a degradative process, its cytoprotective effect on the survival under starvation conditions are widely known, i.e., mitochondrial and cell compounds recycle [11]. Furthermore, studies by Valero et al. (2019) [12] observed that autophagy is required for survival to sulfur dioxide tolerance in yeast. Because autophagy precedes autolysis during aging, and it is essential for cell maintenance and survival under stress conditions [6], the genetic engineering of autophagy-related genes has been proposed by numerous authors [5,13,14,15] in order to accelerate the autolysis and thus, the acquisition of aging-like properties such as foaming properties [16].

Yeast strains compared in this work, *Saccharomyces cerevisiae* P29 and G1 are commonly used in post-fermentative processes, sparkling and sherry-wines elaboration respectively, both involving large aging periods. The nutrient-limited environment to which yeast cells are subjected under these wine making processes, makes it suitable to study the autophagy. During biological aging in sherry-wines, ethanol increases in the medium leading flor yeasts to change their metabolism in order to use this compound as a carbon source. For it, this type of yeast forms a biofilm or velum on wine surface to reach the oxygen and then metabolize the ethanol [17]. While this process has been reported under secondary fermentation conditions in sparkling wine elaboration, no evidence has been reported in flor yeast so far.

This work represents a continuation of a previous research based on studying the apoptosis and autolysis-related proteins and the proteomic response of wine yeasts to CO_2_ overpressure during sparkling wine elaboration [18]. The current study is a first approach focused on the observed changes of the autophagy-related proteins under second fermentation conditions. This was performed in two wine yeast strains, *S. cerevisiae* P29 and G1, in order to analyze their response to CO_2_ overpressure conditions through a proteomic analysis using protein fractionation (OFFGEL) and detection (spectrometer LTQ Orbitrap/HPLC and databases). Proteomic changes observed in autophagy-related proteins as well as those proteins found with high content under typical sparkling wine production conditions, will enlarge the knowledge about this process in yeasts. Furthermore, the study of the genes that codify these proteins may lead to the development of future strategies for the selection of yeast strains with accelerated autolysis through yeast breeding or genetic engineering of second fermentation strains, in this way reducing cost and time of production, as well as acquisition of organoleptic properties.

## 2. Materials and Methods

### 2.1. Yeast Strains and Second Fermentation Conditions

Yeast wine strains, *S. cerevisiae* P29 and G1, used in this work were acclimated during 5 days using a pasteurized must and once high levels of cell concentration (1.5 × 10^8^ cells/mL for P29 and 1 × 10^8^ for G1) and viability (97% for P29 and 90% for G1) were obtained, these were inoculated in bottles with a commercial base wine and 22 g/L of sugar. ‘Tirage’ was carried out and samples by triplicate were collected at different points along the second fermentation: middle of the second fermentation (T1) when pressure reached 3 bar, and one month after it (T2), once 6.5 bar were obtained. Each yeast strain was fermented in two study conditions: sealed bottle (pressure condition or PC) and open bottle (non-pressure condition or NPC). Similar values of ethanol content and sugar consumption were taken into consideration at the time of collecting the samples of control condition. Cultures composition, study conditions and sampling points are described more in detail in Porras-Agüera et al. (2019) [18].

### 2.2. Protein Extraction, Identification, and Quantification

Methods explained in Porras-Agüera et al. (2019) [18] and Moreno-García et al. (2015) [19] were used for protein extraction and identification through LTQ Orbitrap XL (Thermo Fisher Scientific, San José, CA, USA) coupled to a nanoflow LC/MS UltiMate 3000 HPLC system (Dionex, Sunnyvale, CA, USA). As for the protein quantification, protein content (mol%) was calculated following the method explained in Ishihama et al. (2005) [20].

Once identified, those proteins related to autophagy were selected by using the ontology tools provided by SGD (*Saccharomyces* genome database, https://www.yeastgenome.org/), Uniprot (https://www.uniprot.org) databases and literature.

### 2.3. Confidence Criteria and Statistical Analysis

From the total of proteins detected, only those which were obtained with a score > 2 and observed peptides ≥ 2, were selected to proceed with the analysis, in order to provide significant proteins [21,22]. Proteins detected to be over-represented under PC (ratio PC/NPC ≥ 2) were highlighted and discussed in detail. In addition, proteins found specifically in each yeast strain, along with those which reached high protein contents and down-represented under PC (ratio PC/NPC ≤ 0.5) were also considered.

For proteome data, the software Statgraphics Centurion version XVI (StatPoint Technologies, Warrenton, Virginia VA, USA) was used to perform a multiple-sample comparison procedure (MSC), considering a confidence level of 95.0% according to Fisher’s least significant difference (LSD) method, and a multiple variable analysis (MVA), with the aim to distinguish the proteomic response of each strain. The software STRING version 11.0 (available online, https://string-db.org/) was used to create the interaction network map, forming specific protein groups through a MCL (Markov Cluster Algorithm) clustering method. This algorithm accepts a parameter called ‘inflation’ that it is indirectly related with the precision of the clustering. Data were previously normalized through the root square and auto scaling.

## 3. Results and Discussion

In this study, a proteomic analysis was carried out to identify specific proteins involved in the autophagy process and characterize the CO_2_ overpressure effect in two industrial wine yeast strains subjected to second fermentation conditions. To provide a better understanding of the molecular process that involves autophagy, we classified the proteins into different steps: regulation of induction, autophagosome-generating machinery, cargo packaging, vesicle nucleation, vesicle expansion and completion, retrieval, docking and fusion, vesicle breakdown, permease efflux, and mitophagy. A total of 33 autophagy-related proteins were detected in both yeast strains (P29 and G1) and although frequency values remained constant in each condition, CO_2_ overpressure resulted in a considerable decrease of both number of total proteins and protein content from T1 to T2, especially in G1 (Table 1 and Appendix A). The highest values of abundance were obtained in both yeast strains growing under NPCT1. The highest contents were observed in open bottle (NPC), including the regulators at T1 and those proteins involved in cargo packaging at T2 (Appendix A). However, protein contents under PCT1 also highlighted in both strains. Proteins required for vesicle nucleation, expansion, retrieval, docking and fusion, vesicle breakdown, and permease efflux, were not identified at PCT2, and NPCT2 in the case of vesicle nucleation. It could be explained since the number of proteins and content detected at this sampling time in both strains were low (Table 1), which may indicate that autophagy is not relevant once secondary fermentation is over. On the other hand, mitophagy-related proteins were found only under NPC and especially in P29 (Table 1). Results clearly show a detrimental effect of pressure on autophagy-related proteome, particularly in flor yeast. In fact, in recent studies published by our research group [18], a significant decrease was observed in cell viability in both yeast strains under pressure conditions, although it was more remarkable in G1. In addition, the kinetic of second fermentation carried out in both strains through the pressure values, data published also in Porras-Agüera et al. (2019) [18], revealed that P29 is more adapted to pressure conditions and G1 showed a slightly slower kinetic, reaching the maximum pressure (6.5 bar) at 23 and 28 days, respectively.

The effect of pressure on autophagy-related proteins can be better appreciated in Figure 1. According to Figure 1A, CO_2_ overpressure affected more to protein number in flor yeast and especially, to those proteins involved in vesicle nucleation and retrieval. On the contrary, the highest protein number was observed in open bottle (NPC, Figure 1B), mainly in those proteins responsible for regulation, autophagosome formation, and vesicle expansion. Sunray plots based on multivariate data analysis of autophagy proteins detected in both strains, shown in Figure 2, provided differences in protein content means when they were growth under both PC and NPC. According to this analysis, samples taken at T1 differ from those collected at T2, under both conditions. The smallest polygons were found at T2 in both yeast strains, indicating that—apart from the pressure—the second fermentative development also seems to affect negatively to autophagy-related proteome. On the other hand, those samples not subjected to CO_2_ overpressure (NPC) had the most regular polygons and particularly at T1, mainly due to the high amount and content of proteins required for autophagosome and vesicle formation detected in both strains (Figure 1 and Appendix A). In this context, sunray plots obtained from multivariate analysis establish a ‘fingerprinting’ of the autophagy proteome response under each study condition, providing relevant information about the behavior of each wine yeast strain along the prise de mousse.

In order to know the possible interactions between autophagy proteins, a protein–protein interaction network map was built using STRING v11.0 and it is provided in Figure 3. The interaction map showed a high amount of connections between the total autophagy-related proteins (33), represented as nodes, identified in both strains. A total of 177 interactions (number of edges) were observed, with a PPI enrichment *p*-value < 1 × 10^−16^. Such an enrichment indicates that the proteins are at least partially biologically connected as a group. MCL clustering clearly grouped those autophagy proteins involved mainly in regulation and induction (blue nodes), autophagosome formation and vesicle transport (red nodes), and vesicle and vacuole fusion (green nodes). The strongest interactions were observed in those proteins clustered in red color. Only the protein Ald6p showed no interaction with the rest of the proteins, pointing to the fact that this protein is just a cargo and not an active player of the process.

From now on, the autophagy steps as well as the over-represented proteins under PC, specific and those detected with high protein contents in both strains are discussed in depth below.

### 3.1. Regulation of Induction

The most noticeable event among regulators was the detection of the proteins Bcy1p and Sec13p (Figure 4). Sec13p was detected 3.2-fold under PCT1 in G1 (Table 2) and, although it is part of COPII vesicles, required for the proper transport of proteins from endoplasmic reticulum (ER) to the Golgi [23], studies by Panchaud and Péli-Gulli (2013) [24] demonstrated that it is involved also in the regulation of TORC1 complex through the interaction with a GTPase activator. Besides this regulator, the regulatory subunit of the cyclic AMP-dependent protein kinase (PKA) Bcy1p, known for negatively regulating autophagy [25], was found specifically in P29 under PCT2 (Table 2).

The phosphatase 2A complex (PP2A) subunits, Pph21p and Pph22p, decreased more their content in P29 than in flor yeast, especially at PCT1 (Figure 4). Studies have reported a role in autophagy regulation via TORC1 interaction [26]. Although Sec13p participates in vesicle trafficking as component of COPII vesicles, it has been recently reported a link between secretory pathway and autophagy [27] in which COPII vesicles fuse with Atg9p vesicles to provide membrane source and regulate the autophagosome abundance.

### 3.2. Autophagosome-Generating Machinery

Proteins involved in autophagosome formation were detected in both yeast strains and most of them were found down-represented in P29 strain (Figure 4A). Among the over-represented proteins under PC we can stand out: Shp1p (3-fold under PC in G1) and Sec18p (Table 2). The first one showed the highest content increase value in the case of flor yeast G1 at T1 and in P29 at T2 where it was specific (Figure 4). Studies by Krick et al. (2010) [28] demonstrated that Shp1p is also essential for autophagosome biogenesis, via interacting with Atg8p (Figure 3) through a system that allows to detect protein interaction in medium without uracil. In fact, ubiquitin-like protein Atg8p, was down-represented at NPCT1 (both in P29 and G1) and T2 (only in P29) (Figure 4). Since this protein is required to form autophagosomes during starvation conditions [29], it is acceptable to think that Atg8p also participates in vesicle formation during Cvt pathway. Huang et al. (2000) [30] used an atg8 mutant strain and confirmed its role in both pathway during starvation conditions due to the inhibition of prApe1p import. Apart from Shp1p, the chaperone Sec18p was found highly represented under PCT1 only in the flor yeast G1 (Table 2). Its presence could be explained due to its role in autophagosome formation and the fusion with the vacuole [31]. Moreover, these results agree with those obtained by Penacho et al. (2012) [32], where genes involved in vacuolar functions were reported to be overexpressed under second fermentation conditions.

As for the rest of autophagosome formation proteins, most of the Atg proteins appeared to be down-represented under PC: Atg2p, Atg3p, Atg4p, Atg9p, Atg18p (autophagy core machinery), Atg21p and Atg27p (specific of Cvt pathway). All decreased their content in P29 under PC and all, except Atg21p and Atg27p (T2), were observed at T1 (Figure 4 and Appendix A). The interaction between Atg proteins, such as Atg2p, Atg18p, or Atg9p shown in Figure 3, has been reported by several authors and is essential for correct autophagy process [33,34]. These results seem to indicate that autophagy takes place at T1 in both strains and under both conditions, once nutrient levels drop in the wine, due to the higher amount of autophagy-related proteins detected in respect to T2. However, since Atg-related proteins were found with low content and even most of them were not identified under pressure, it suggests a possible negative effect of pressure on autophagy-related proteome.

### 3.3. Cargo Packaging

Yeast autophagy (and Cvt pathway) involves the transport of hydrolases enzymes, Ape1p (aminopeptidase I) and Ams1p (α-mannosidase) into vesicles and delivery to the vacuole. In this context, both the vacuolar aminopeptidase Ape1p, often used as a marker protein in studies of autophagy and Cvt pathway [35], and the mannosidase Ams1p, were identified as over-represented under PCT1 in G1 and specific under PCT2 in P29, respectively (Table 2). Ams1p is delivered to the vacuole in a novel pathway separate from the secretory pathway and requires the Cvt and autophagy components [36]. The presence of both enzymes overrepresented under PC suggests that they are being delivered to the vacuole for degradation of organelles and cell compounds in both strains.

As for the rest of the proteins involved in cargo packaging, the protein content of Ald6p or aldehyde dehydrogenase was highlighted under both conditions. This was only detected under NPC, showing a considerable decrease under PC in both strains (Figure 4). Studies by Onodera and Oshumi (2004) [37] demonstrated that in addition to Ams1p and Ape1p, the protein Ald6 is also specifically targeted to the vacuole by autophagosomes under nutrient starvation conditions, and it was quickly depleted in cells as a result of a preferential degradation of this protein during autophagy. Consequently, this depletion has been used as a marker for the autophagy process [6]. According to the protein content obtained under both conditions, it might be suggested that the autophagy process occurs when yeast cells are subjected to pressure conditions, representing the first evidence of this process in flor yeast, as it has not been reported yet. However, since this protein is a key player in the conversion of acetaldehyde to acetyl-CoA during growth on non-fermentable carbon sources such as acetaldehyde or ethanol [38] it might indicate a possible role in gluconeogenesis, especially at T2.

### 3.4. Vesicle Nucleation

Autophagic vesicles are constructed at the PAS from newly generated membranes, and the formation of the core and the new membrane require the participation of Atg9p and the phosphatidylinositol 3-kinase (Ptdlns3K) complex I, which includes the Ptdlns 3-kinase Vps15p, Vps30p, Vps34p, and Atg14p [39]. Protein content in this category was reported with the lowest levels in both yeast strains and most of the proteins were detected especially at T1 under both conditions (Figure 4). Under PC, only Vps15p was found over-represented at T1 in P29 (Table 2). This protein, together with Vps30p (down-represented under PCT1 in G1), have been associated with both autophagy and carboxypeptidase Y sorting [40]. On the other hand, Atg9p, transmembrane protein involved in forming Cvt and autophagic vesicles, was detected only under NPCT1 in P29 (Appendix A).

### 3.5. Vesicle Expansion and Completion

Proteins involved in this autophagy step are two ubiquitin-like conjugation system (Atg12 and Atg8 systems), Sec2/4p, Ypt1p, and complexes COG and TRAPPIII. Most of these proteins were down-represented at PCT1 in both strains (Figure 4). The guanine nucleotide exchange factor Sec2p was the only over-represented protein under PC (3.4-fold) at T1 in P29 (Table 2). Studies by Geng et al. (2010) [41] demonstrated that this protein, after autophagy induction, participates in autophagosome formation. Apart from the detection of Sec2p, the RabGTPase Ypt1p was identified with the highest content at PCT1 in flor yeast; however, it was not detected at T2 (Appendix A). This protein is required for vesicle docking and targeting during ER to Golgi trafficking, and also is involved in autophagy regulation participating in PAS formation and assembly [42].

### 3.6. Retrieval

The proteins that participate in retrieval of PAS were reported with low protein contents and most of them decreased their content under PC (Figure 4). Among them, Atg27p, involved in membrane delivery to the PAS and required for both autophagy (autophagosome assembly) and Cvt pathway [43], decreased the content under PCT1 (Figure 4). Atg27p shuttles between the mitochondria, PAS, and the Golgi complex. In addition, it participates in anterograde transport of Atg9p from the mitochondria to the PAS [44]. The anterograde cycling to the PAS requires Atg9p, detected in both strains and down-represented only under PCT1 in P29, Atg11p (found at T1 in both conditions but more under PC) and Atg23p (not detected), while retrograde cycling from the PAS to the mitochondria or Golgi complex involves the Atg1p-Atg13p complex, Atg2p and Atg18p (all of them detected except Atg13p). Atg18p (down-represented only at T1 in both strains) has been reported as essential for vesicle formation in both autophagy and Cvt pathway [45].

### 3.7. Docking and Fusion

Once the autophagosome is formed, it releases the content by fusion with the vacuole. SNARE proteins (Vam3p, Vam7p, Vit1p, and Ykt6p), Rab GTPases such as Ypt7p, the chaperone Sec18p and Vps proteins participate in this process [46]. In general, protein content values were low in both conditions and strains, although it was more pronounced in G1 (Appendix A). In P29 two proteins, Ypt7p and Ykt6p, required for fusion events, not autophagic in the case of Ykt6p [47,48], were found to be specific under PCT1 (Table 2). The association of Ypt7p with Vps complex is required for vacuolar fusion. Seals et al. (2000) [49] observed that interaction between Ypt7p and Vps proteins (Figure 3) is required for an efficient vacuolar fusion in yeasts, something that—considering our results—may take place at T1 in both strains. Among the components of this Vps complex (Vps8p, Vps16p, and Vps41p), only the last one was found specifically in G1 (Figure 4B).

### 3.8. Vesicle Breakdown and Permease Efflux

After fusion to the vacuole, two conserved components are involved in breakdown of the autophagosome and permease efflux in yeasts, Atg15p and Atg22; however, none were detected in this study.

### 3.9. Mitophagy

Mitophagy in yeast can be induced under starvation conditions, oxidative stress, and in nonfermentable mediums, representing a selective autophagy process in which a mitochondrion is degraded by macroautophagy [50]. In fact, most of Atg proteins are required for this process [51]. Proteins involved in mitophagy such as Atg32p and Dnm1p were identified only in P29 under NPCT1 (Appendix A). During mitophagy, Atg32p is essential to initiate the process recruiting the adaptor Atg11p and the ubiquitin-like protein Atg8p [52]. The presence of these proteins suggests that P29 might be performing a mitophagy process, probably as an adaptive mechanism to survive under starvation conditions or even to protect against oxidative stress, thus removing the ROS [53], under non-pressure conditions.

## 4. Conclusions

This work represents a first approach based on the identification and comparison of autophagy-related proteome in two industrial wine yeast strains commonly used in post-fermentative processes, under pressure conditions. According to proteomic results, autophagy seems to take place during the fermentative stage in the both yeast strains. Furthermore, CO_2_ overpressure affects negatively to autophagy proteome in terms of protein number and content in both strains, although this effect was more remarkable in the flor yeast. Under pressure conditions, regulators, and proteins related to autophagosome formation highlighted in flor yeast, while those involved in vesicle nucleation and expansion were more relevant in sparkling wine yeast strain. Apart from contributing to the knowledge about yeast autophagy, those specific and highly represented proteins under second fermentation conditions—such as Bcy1p, Sec2p, Sec13p, Sec18p, Shp1p and Vps15p—could be used as biomarkers for accelerating the autolysis during aging period in sparkling wine elaboration. The study of the genes that codify these proteins would allow promote autolysis in wine yeasts through genetic engineering, thus reducing cost and time production, as well as the acquisition of good organoleptic properties. Moreover, this work opens the door to the use of flor yeasts for sparkling wine elaboration. However, further research, including different approaches and disciplines such as genomics and metabolomics, along with studies focused on protein activity and electron microscopy imaging, is needed to achieve more solid conclusions.

## Figures and Tables

**Figure 1 microorganisms-08-00523-f001:**
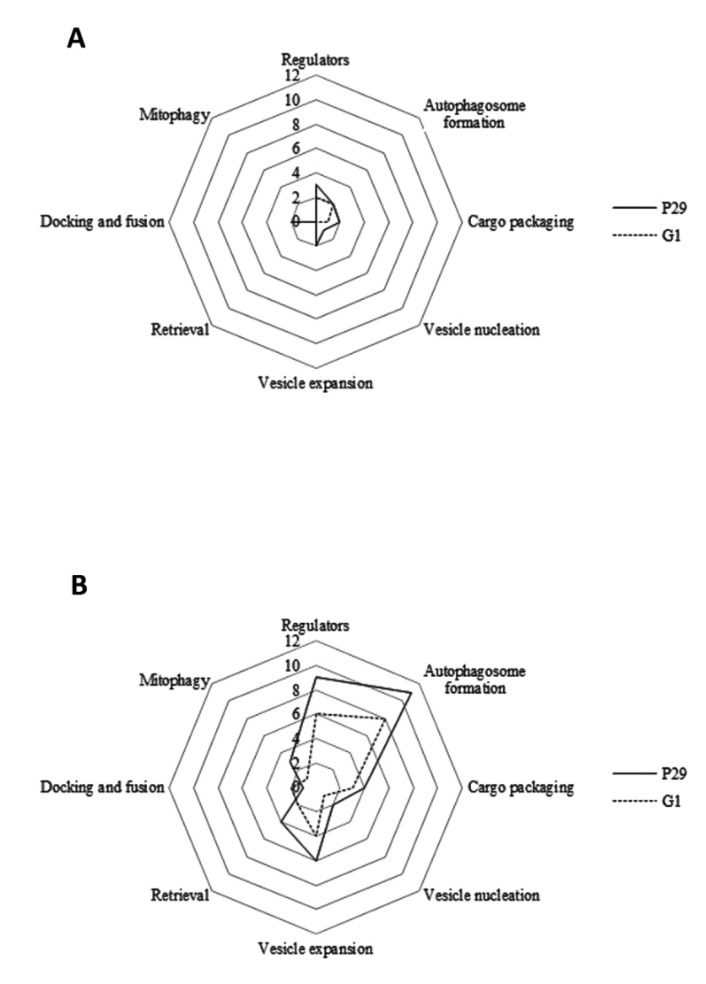
Total number of autophagy-related proteins required for each process step identified in *S. cerevisiae* P29 and G1 under (**A**) PC (endogenous CO_2_ overpressure condition) and (**B**) NPC (non-pressure condition).

**Figure 2 microorganisms-08-00523-f002:**
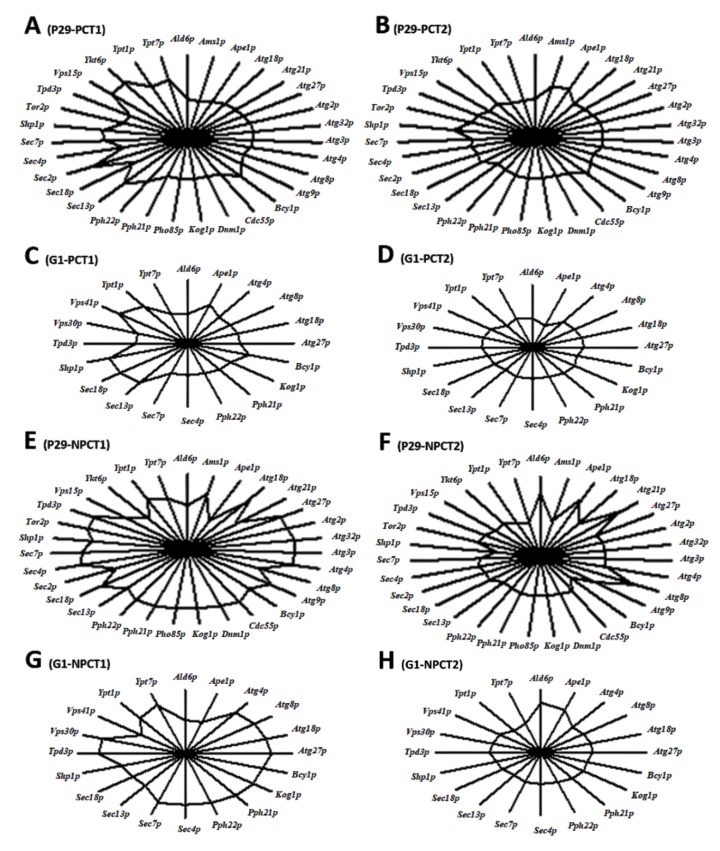
Sunray plots obtained by multivariate data analysis of autophagy proteins detected in *S. cerevisiae* P29 and G1. Each ray represents a protein and the distance from the center to each vertex indicates the value of each protein. The end of the ray corresponds to the mean value plus three standard deviations and the center the mean minus three standard deviations. (**A**) P29-PCT1; (**B**) P29-PCT1; (**C**) G1-PCT1; (**D**) G1-PCT2; (**E**) P29-NPCT1; (**F**) P29-NPCT2; (**G**) G1-NPCT1; (**H**) G1-NPCT1. PC (endogenous CO_2_ overpressure condition), NPC (non-pressure condition), T1 (middle of the second fermentation), T2 (one month after the second fermentation).

**Figure 3 microorganisms-08-00523-f003:**
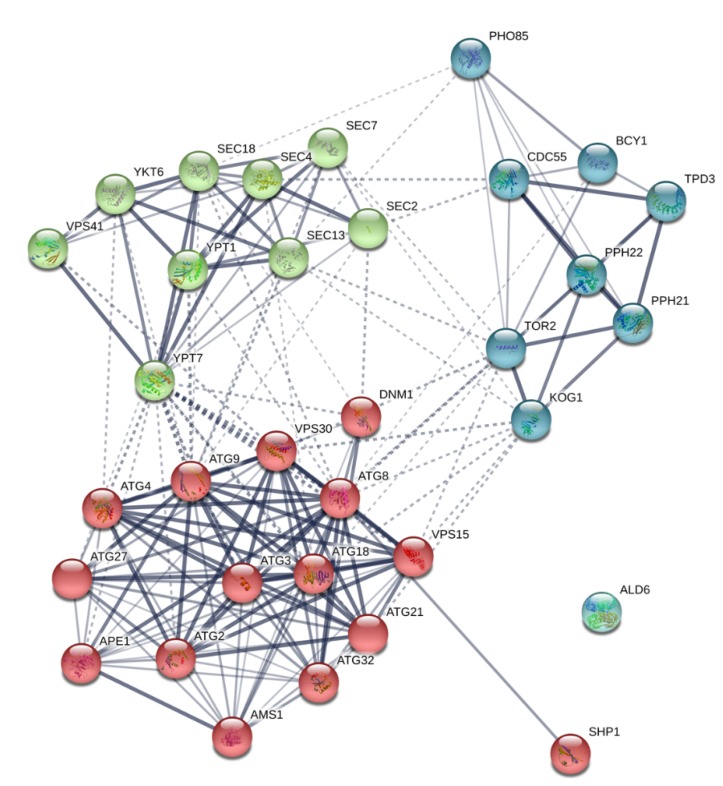
Interaction network map built using STRING v11.0 and based on the 33 autophagy-related proteins in total detected in *S. cerevisiae* P29 and G1. Proteins are showed as nodes and the existence of interactions between them are represented by lines (connection between nodes). Line thickness indicates the strength of the different interactions. Nodes with the same color represent specific clusters: autophagy regulation and induction (blue nodes), autophagosome formation and vesicle transport (red nodes), and vesicle and vacuole fusion (green nodes). PPI enrichment *p*-value < 1 × 10^−16^.

**Figure 4 microorganisms-08-00523-f004:**
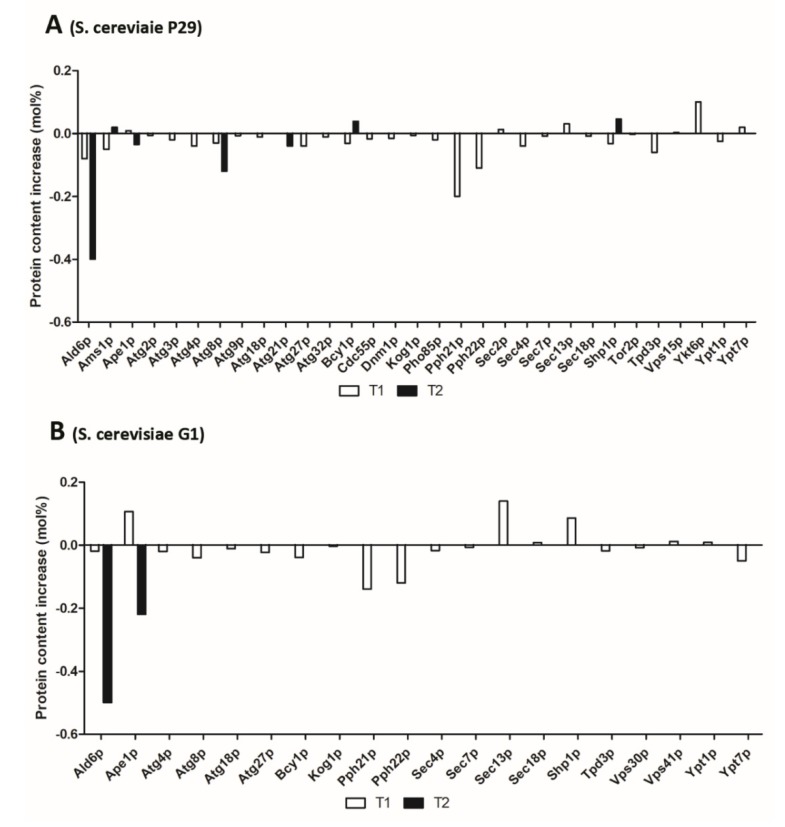
Protein content increases (mol%) under PC (pressure condition) of autophagy-related proteins compared to NPC (non-pressure condition). (**A**) protein content increases identified in *S. cerevisiae* P29. (**B**) protein content increases identified in *S. cerevisiae* G1. T1 (middle of the second fermentation), T2 (one month after the second fermentation).

**Table 1 microorganisms-08-00523-t001:** Frequency of the autophagy-related proteins identified in both yeast strains (*S. cerevisiae* P29 and G1) under PC (pressure condition) and NPC (non-pressure condition), at the middle of the secondary fermentation (T1) and one month after it (T2)

	^a^ Protein Frequency in *Saccharomyces cerevisiae*	PC T1		NPCT1		PCT2		NPCT2	
		Protein Frequency		Protein Frequency		Protein Frequency		Protein Frequency	
		P29	G1	P29	G1	P29	G1	P29	G1
Total proteins	94 out of 6721, 1.4%	11 out of 594, 1.85%	7 out of 568, 1.23%	29 out of 1517, 1.91%	19 out of 1000, 1.90%	4 out of 419, 1.33%	−	4 out of 392, 1.86%	2 out of 218, 0.91%
Regulation of induction	26 out of 6721, 0.39%	3 out of 594, 0.51%	2 out of 568, 0.35%	9 out of 1517, 0.59%	6 out of 1000, 0.60%	1 out of 419, 0.24%	-	-	-
Autophagosome-generating machinery	24 out of 6721, 0.36%	2 out of 594, 0.34%	2 out of 568, 0.35%	10 out of 1517, 0.66%	8 out of 1000, 0.80%	1 out of 419, 0.24%	-	2 out of 392, 0.51%	-
Cargo packaging	8 out of 6721, 0.12%	1 out of 594, 0.17%	1 out of 568, 0.18%	4 out of 1517, 0.26%	3 out of 1000, 0.30%	2 out of 419, 0.48%	−	2 out of 392, 0.51%	2 out of 218, 0.92%
Vesicle nucleation	5 out of 6721, 0.07%	1 out of 594, 0.17%	-	2 out of 1517, 0.13%	1 out of 1000, 0.10%	-	-	-	-
Vesicle expansion	26 out of 6721, 0.39%	2 out of 594, 0.34%	1 out of 568, 0.18%	6 out of 1517, 0.40%	4 out of 1000, 0.40%	-	-	-	-
Retrieval	7 out of 6721, 0.10%	-	-	4 out of 1517, 0.26%	2 out of 1000, 0.20%	-	-	-	-
Docking and fusion	14 out of 6721, 0.21%	2 out of 594, 0.34%	1 out of 568, 0.18%	1 out of 1517, 0.07%	1 out of 1000, 0.10%	-	-	-	-
Vesicle breakdown	1 out of 6721, 0.01%	-	-	-	-	-	-	-	-
Permease efflux	1 out of 6721, 0.01%	-	-	-	-	-	-	-	-
Mitophagy	10 out of 6721, 0.15%	−	−	3 out of 1517, 0.20%	1 out of 1000, 0.10%	−	−	1 out of 392, 0.26%	−

^a^ The total number of autophagy-related proteins identified until date in *S. cerevisiae* have been included in the first column.

**Table 2 microorganisms-08-00523-t002:** List of over-represented autophagy-related proteins under PC (pressure condition) detected at the middle of the second fermentation (T1) and one month after it (T2), in both yeast strains (*S. cerevisiae* P29 and G1). Proteins specifically found under PC and fold changes of the protein content PC/NPC are shown in brackets.

Yeast Strains	*S. cerevisiae* P29		*S. cerevisiae* G1	
Sampling times	T1	T2	T1	T2
Regulators/inductors	-	Bcy1p (Specific, 0.04)	Sec13p (3.21)	-
Autophagosome-generating machinery	-	Shp1p (Specific, 0.05)	Sec18p (3.40), Shp1p (3.08)	-
Cargo packaging	-	Ams1p (Specific, 0.02)	Ape1p (2.90)	-
Vesicle nucleation	Vps15p (2.27)	-	-	-
Vescicle expansion	Sec2p (3.43)	-	-	-
Retrieval	-	-	-	-
Docking and fusion	Ykt6p (Specific, 0.10), Ypt7p (Specific, 0.11)	-	-	-

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
