# Peer review of "Autophagic Proteome in Two Saccharomyces cerevisiae Strains during Second Fermentation for Sparkling Wine Elaboration"

_microorganisms, 2020, doi:10.3390/microorganisms8040523_

Round 1

Reviewer 1 Report

In their recent work the authors have identified proteins implicated in the yeast autophagy process during sparkling wine elaboration. These proteins have been classified and partially analyzed according to their function and role in the autophagy process. Two S. cerevisiae strains (P29 and G1), under pressure condition and non-pressure condition, have been tested for two moments during the second fermentation inside the bottle. The present work is the continuation of a previous published paper, entitled ‘’ First Proteomic Approach to Identify Cell Death Biomarkers in Wine Yeasts during Sparkling Wine Production’’. The core of the research (all the fermentation trials, proteome analysis focused on the apoptosis and autolysis-related proteins) has already been published.  

 As they claim in the Introduction (line 66-67-68) ‘’The current study focuses on the changes experimented at proteome level associated with other relevant process which occur under second fermentation conditions as autophagy’’. According to my opinion the aim of their work is not well documented. What they mean by changed experimented? Maybe it would be nice to mention more the interest of these tested proteins to foam and wine quality. Why is it that interesting to study them?

Additionally, they claim that the corresponding results may allow to propose strategies for the selection of yeast strains with accelerated autolysis through yeast breeding or genetic engineering of second fermentation strains, reducing in this way cost and time production (line 73-74-75). How that can happen and how these statements are correlated with their results? As the results are presented I can’t see any correlation or proposed strategy.

Even if the present work is well written and the scientific level is high, I believe that it can’t support an independent published work. It would be interesting to study the effect of these detected proteins related to autophagy to a technological characteristics i.e. to foam formation and quality.

Some more detailed comments to the manuscript follow:

Line 24: you should add here that the flor yeast is G1 and that this yeast is implicated in sherry elaboration

Line 40: more references are needed for this statement

Line 42: proposed instead of purposed

Line 73-75: Can you explain more the link between your results and this general statement?

Line 79: ‘’high level of cell concentration’’, specify the population level please

Line 97-98: any bibliographic reference for you chosen confidence criteria?

Line 121: What is T1 and T2? It is explained only in the table

Line 338: Can you explain more this assumption

Figure 4: The increase (or decrease) is compared to what?

Line 247-248: Can you explain more your indication please?

Reviewer 2 Report

Comments to the Author

I read this manuscript with interest and I believe this manuscript must publish as possible as faster.

However, I think that the author needs a little revision to accept by Microorganisms.

In my personal opinion, I felt that you can submit this manuscript to a better journal like Food Microbiology.

Minor comments

  1. Figure 2:

In this figure, the characters on the figure are unclear and difficult to understand.

  1. Table 2:

Please fill the expression level of Shp1p in the Table2.

Round 2

Reviewer 1 Report

Dear authors,

the manuscript has been ameliorated a lot. I am completely covered by your answers. I agree for publication